# How Does Mission Statement Relate to the Pursuit of Food Safety Certification by Food Companies?

**DOI:** 10.3390/ijerph17134735

**Published:** 2020-07-01

**Authors:** Quan Lin, Yutao Zhu, Yue Zhang

**Affiliations:** 1Department of Business Administration, Business School, Shantou University, Guangdong 515063, China; 17ytzhu@stu.edu.cn; 2Department of Accounting and Information Systems, David Nazarian College of Business and Economics, California State University, Northridge, CA 91330-8372, USA

**Keywords:** food safety certification, mission statement, organizational legitimacy, SOE, majority shareholder control

## Abstract

Food safety has long been a major public concern in China. One question of the food processing industry’s emphasis on food safety social responsibility is whether a food processing company should pursue food safety certification for its products. As part of their corporate image, some food processing companies focus on food safety in their corporate mission statements. To enhance the legitimacy of a mission statement, as a guide for a firm, can provide food companies the legitimacy of perhaps pursuing food safety certification. However, we find that under different equity natures, the pressures on the normative legitimacy of the firm are different and the impact of mission statements on the acquisition of food safety certifications is also different. By analyzing the mission statement of companies in the Chinese food industry, we find that firms with a mission focusing on food safety concerns are more willing to pursue food safety certification. Moreover, compared to the firms with more distributed shareholder ownership, in firms where a majority shareholder has substantial control, the relationship between mission statements and the possession of food safety certification is stronger; compared to non-state-owned enterprises, in state-owned enterprise (SOEs), the relationship between firm mission statements of and the acquisition of food safety certification is stronger.

## 1. Introduction

Since the 1980s, the concept of corporate social responsibility (CSR) has attracted more and more attention worldwide. With globalization and China’s accession to the WTO, CSR is rapidly spreading across China and is receiving increasing attention. Food production is the foundation of human survival and well-being; food safety also involves critical common interests of the public and concerns the health or even life of every member of society. Therefore, food processing companies (hereafter “food companies”) must take the initiative to assume corresponding social responsibilities in food safety. In today’s China, however, increasingly frequent food safety incidents pose great dangers—potential or current—to the safety of the public. The shocking Sanlu (“Three Deers”) milk powder incident in 2008 (where China dairy companies added melamine instead of protein to baby milk formulas) seriously undermined consumer confidence in “well-known companies” and “inspection-free food products”. In the face of the resulting public uproar, the Chinese government escalated its regulation as well as the enforcement of food-safety-related laws and regulations. Both the public and the government are eager to find mechanisms to improve and secure food safety. More and better understanding of mechanisms internal to food processing companies, including the role of CSR on food safety promises and fulfillment of such responsibility, is in need.

The issues of food safety and CSR are studied by many scholars based on the agency, institutional, and neo-institutional theories [1,2]. The agency theory aims at identifying an optimal contract that can coordinate the divergent interests of a principal-agent relationship [3,4]. Considering the impact of the complexity of the organization on the environment, it only proposes the accountability mechanism between the agent and the principal [5] and does not stand on the main body of organizational sociology. The behavior of food companies needs to abide by the laws and regulations and be consistent with social norms or cultural cognitions in the institutional environment in which the companies operate. This behavior is very important for firms to obtain resources and legitimacy. Given this background, we believe that food safety is an essential factor for the survival of food companies.

Bhattacharya and Sen [6] believe that environmental pressures (such as legal environment or market environment) encourage companies to fulfill their social responsibilities. Faced with dual pressures of growth (in outputs or profit) and societal demands, food processing companies can struggle to devote resources to competing goals. We can find other ways inside the company to help food processing companies to consciously fulfill their social responsibilities for food safety. The mission statement of a company, as a shared value and behavioral code for all members of the company, can have broad and profound impact on the belief, pursuits, and behavior of members of the company [7,8,9]. Most of the existing studies have recognized the value of mission statements for financial performance [10,11], but few researchers have studied the contents of missions from the perspective of social responsibility. In general, management and corporate stakeholders are very concerned about the performance of the business as measured by “key performance indicators” (KPIs), such as profit, cost, market share, so those missions that focus on KPIs play a more visible or influential role. Unfortunately, not all shareholders or managers are sufficiently concerned about corporate social responsibility for food safety. The focus on food safety in the mission statement of a food company may result from the company’s pursuit of a positive image in response to government and public expectations, or from the interests of the company shareholders [12,13,14]. Moreover, the mission statements of Chinese companies are society-oriented and tend to emphasize the social roles of an organization [15], so will those food companies with a high focus on food safety in their mission statement be more willing to undertake more direct and substantial actions regarding food safety, often reflected in the pursuit of food safety certification? Under what circumstances can a food safety-focused mission statement be effective? The following briefly summarizes existent studies on the effectiveness of mission statements’ affecting behaviors of an organization.

According to Bartkus, et al. [16], an effective mission statement can affect the behavior of firms. The effectiveness of mission statements has been studied by researchers and the following three aspects of effective mission statements have been identified: (1) the mission statement includes specific, relevant components [9,17,18,19]; (2) the mission statement clearly refers to key stakeholders [17]; and (3) the mission statement clearly highlights the objectives of the organization [20]. While the first two aspects of effective mission statements have been more thoroughly studied, the third aspect (“mission statements clearly highlighting the objectives of the organization are more effective”) remains to be further examined. We will follow this line of study [9,16,21] to conduct our examination of the topic. Moreover, based on organizational legitimacy theory [2,22,23], we provide a detailed examination into the mission statement’s effect on a food company’s pursuit of food safety certification.

### 1.1. Theory and Hypothesis

#### 1.1.1. The Relationship between Mission Statement and Food Safety Certification

In the studies on the issue of food safety, one of the theories most frequently employed is that of organizational legitimacy [2,22,23]. Organizational legitimacy means that the actions taken by an organization are “desirable, proper, or appropriate within some socially constructed system of norms, value, beliefs, and definitions” [24]. Singh, et al. [25] categorized organizational legitimacy into internal legitimacy and external legitimacy. Internal legitimacy refers to the recognition, support, and obedience of the members of the organization to the authority structure, with the role of strengthening organizational practices and motivating members to act in shared ethical, strategic, or ideological contexts [26]. External legitimacy of an organization comes from the support to and recognition of the organization from the government, regulatory agencies, and external stakeholders [27]. External legitimacy is rooted in a wide range of social realities and relies on regulatory agencies, organizational boundaries, and reputational norms.

Legitimacy is obtained when an organization’s behavior is consistent with laws, regulations, social norms, or cultural cognitions in an institutional environment. An organization will face pressure of internal and external legitimacy when its behavior is inconsistent with or has significant deviations from its institutional environment’s expectations. Moreover, realistic factors such as the complexity of the institutional environment, the scarcity of resources, and the limited availability of resources make it difficult to simultaneously address internal and external pressures [28].

The behavior of firms in fulfilling their social responsibility can provide them behavioral legality in the institutional environment of system, norms, cultural concepts, and enhance the firms’ ability to survive and thrive in a complex environment. This behavior signals its stakeholders that the company has made substantial efforts to improve social benefits and social welfare [24]. Through cultural cognitive legitimacy, food processing companies establish behavioral logic for their fulfilment of social responsibilities in the arena of food safety. In modern society, structures of organizations in a certain field or industry tend to converge [29]; this converge in the context of food processing industry forms the collective behavioral logic for food companies, and thus provides the internal legitimacy for the food companies.

First, the implementation of food safety social responsibility is under the impact of cultural-cognitive legitimacy. A mission statement, as the future direction for and the cultural symbol of the company [30], is an essential source for the corporate social responsibility cultural-cognitive legality. The mission statement forms a “cultural glue” by integrating and interpreting the values, codes of conduct, objectives, and business ideas of the company, which can enhance the awareness of the members of the public and other enterprises [7]. The organization publicly commits to achieving specific goals and ideas, uses mission statements to communicate and establish the organization’s image, and at the same time discloses the organization’s commitment to external stakeholders [11,31]. In this situation, corporate behaviors consistent with mission statements, in this case the focus on food safety in mission statement, will be supported and approved by the public.

Second, the implementation of food safety social responsibility is under the impact of internal legitimacy. The mission statement is essential for the effective strategic management of a company. It is a crucial starting point for effective and successful strategic management. A well-documented mission statement is a vital element in an effective strategic planning process [32,33]. Meanwhile, mission statements can also guide the behavior of the employees [20]. It conveys the information what the firm will do, endows the decision-maker with legitimacy and helps him or her to facilitate the implementation of decisions made.

**Hypothesis** **1.**
*The concern about food safety issues in the mission statement is positively related to the possession of food safety certification by the food companies.*


#### 1.1.2. The Moderate Effect of the Control of the Largest Shareholder

In China, the equity of publicly listed companies is usually relatively concentrated: the ownership and control rights tend to be concentrated in the hands of one or a few large shareholders. The management of the company is usually directly assigned by the largest shareholders, and the values and management philosophy of the management and the largest shareholders tend to be consistent [34]. This mechanism allows these largest shareholders to influence the organization’s objectives and management’s control [35]. The legitimacy of corporate behavior stems from the consistency of the goals of management and the interests of the majority shareholders. In the situation of frequent food safety incidents, it is difficult for food companies to secure the trust of consumers. Therefore, to improve the food companies’ image of good corporate citizens, the majority shareholders may be more willing to take on more food safety responsibilities [13,14].

The equity nature or the concentration of equity of a food processing company as discussed above would affect the food company’s possession of food safety certification. The higher the largest shareholder’s shareholding ratio, the more influence the largest shareholder will have on the board of directors and senior management of the company. Because of this influence, board members and senior managers of the company may act as the trustees of the largest shareholder [36]. In enterprises with a large holding by the largest shareholder(s), since the management has higher level of consistency with the largest shareholder in values and management philosophy, there tends to be less conflict between the two (management and largest shareholder). Mission statement is the document that explicitly and prominently reflects the values and management philosophy of largest shareholder and the management; the consistency of the two parties (largest shareholder and the management) gives mission statement higher level of legality. In this situation (when the largest shareholder and the management agree to each other more), there is a higher level of legality embedded in the mission statements. With this higher legality embedded, when the mission statement focuses on food safety, it can exert larger legitimate power, or the management would form strategies and policies according to such a mission statement, which could lead to commitments to food safety and to a successful pursuit of food safety certification resulting in the possession of a certificate.

To sum up, the higher the shareholding of the largest shareholder, the greater the impact of the mission statement on the company’s important decision-making and management actions, such as pursuit of food safety certification. We thus have the following hypothesis:

**Hypothesis** **2.**
*In firms with a higher share percentage of the largest shareholder, the positive correlation between mission statement’s food safety concerns and the possession of food safety certification will be strengthened compared to firms with more distributed shareholder ownership.*


#### 1.1.3. The Moderate Effect of the Nature of the Ownership

In state-owned enterprises (SOEs), the largest shareholder is the government, and management is also appointed by the government. The governing structure and the management structure, thus the mission statement of SOEs, more directly reflect the will of the government. There is a natural link between the SOEs’ mission and government policies. As a representative of the public interest, the government bears the liability of maintaining social responsibility for food safety [37]. This leads to the business objectives of SOEs having both economic and social attributes: the SOEs participate in market competition to achieve the goal of enterprise value-adding, and they also bear a wide range of social responsibilities [38]. This means that SOEs are more inclined to respond to government policies and serve as role models for other companies [39,40]. Therefore, the mission statement of SOEs often reflects government policies. For state-owned food processing companies, food safety is a high-priority objective. The SOEs thus concern more about food safety in their mission statement. The management would form strategies and policies according to such a mission statement, which could lead to commitments to food safety and to a successful pursuit of food safety certification resulting in the possession of a certificate.

**Hypothesis** **3.**
*In state-owned enterprises, the positive correlation between food safety concerns and the possession of food safety certification will be strengthened compared to non-state-owned enterprises.*


Figure 1 summarizes the framework of our study.

## 2. Materials and Methods

### 2.1. Sample

The sample in our study consists of all Chinese A-share (Chinese domestic shares denominated in Renminbi or China Yuan—and traded on the Shanghai and Shenzhen stock exchanges) listed firms in the food industry (e.g., agro-food processing industry, food manufacturing, and wine, beverage and refined tea manufacturing) in the years from 2009 to 2018. Our sample year starts from 2009 when the State Food and Drug Administration (SFDA), the Chinese quality and food safety watchdog, cancelled all kinds of national inspection exemptions previously given to food producers. From the total population of firms, we excluded those flagged as Special Treatment (ST) and *ST firms from the finance industry and firms with missing information. The stocks flagged as ST and *ST are those who have suffered serious losses for multiple years (two years for ST and three years for *ST) or have significant uncertainties in the firms’ operations. The companies with these stock designations are essentially insolvent and at risk of being delisted. The final sample consisted of 115 firms and 872 firm-year observations. All the financial data are collected from the Chinese Stock Market and Accounting Research (CSMAR) database, the food safety certifications from China National Certification and Accreditation Information Public Service Platform, and the nature of ownership data from the WIND database, a major database on Chinese capital markets.

The China Securities Regulatory Commission (CSRC), the main regulator of the securities industry in China, requires that listed companies‘ annual reports must disclose the company’s basic conditions, stocks and debts, controlling shareholders and actual controllers, management discussions and analysis, etc. Although CSRC does not require the company to disclose corporate mission statement information, the importance of mission statements was promoted by the Forum of Chinese Management Thoughts jointly launched in 2001 by Tsinghua University, Peking University and several other leading universities and consulting firms. The Forum is committed to helping Chinese business executives to simplify complex management theories and localize western corporate cultural values. The text of the mission statement we used was collected through public channels such as the company’s official website and the company’s annual report in October 2019. We followed the methods used by Bartkus and Glassman [11] to study corporate mission statements: looking up the company’s mission statement from the company’s annual report and official website. Following David [18], the mission statement text we collected covers mission, vision, core values, goals, strategy, and principles of business operations.

#### 2.1.1. Independent Variables

We read the text of each company’s mission statement to find out whether it mentions words or phrases related to food safety issues, words/phrases such as “quality”, “improving product quality”, “providing safe products”, “create safety food for consumers”, “production according to national regulations” and so on. All these words and phrases are considered related to “food safety issues”. Based on these words and phrases in food safety vocabulary, this study extracts food safety-related content from corporate mission statement texts. The text parsing and extracting were done with the open source software R Project (License GNU GPL v2, Free Software Foundation, Boston, MH, USA). To eliminate the biased impact of the length of the corporate mission statement text on the length of food safety-related texts, and to ensure the comparability of food safety concerns between different companies, this study adopts the proportion of the focus of food safety in the mission statement to measure food safety concern of mission statement, or, to “normalize” the proportion of food safety contents in the entirety of the mission statement, regardless of the total length of the mission statement. The proportion of food safety contents is defined as follows:(1)$$food\;safety\;concern\;of\;mission\;statement=\frac{Times\;of\;\mathrm{food}\;\mathrm{safety}\;\mathrm{issues}\;semantic\;entries\;in\;the\;mission\;statement}{Number\;of\;all\;semantic\;entries\;in\;the\;mission\;statement}$$

#### 2.1.2. Dependent Variables

To an organization nowadays, one of the common and effective leverages for operational and managerial changes is the pursuit of external certifications. External certifications have the effects of changing the logic of collective action and making enterprises take collective responsibility more seriously [41]. For example, in the environmental protection arena, when a company obtains green certification, it is equivalent to joining a prestigious club [42,43]. Independent third-party inspections increase corporate green behavior responsibility, reduce opportunistic behaviors, and enable companies to operate more actively in accordance with the certification’s standard [43].

Applying the same logic to the issue of our major concern—food safety—Chinese food companies have the following food safety-related certifications: ISO 9000 quality management system certification, ISO 22000 food safety management system certification, and food quality and safety management system certification based on Hazard Analysis and Critical Control Points (HACCP), etc. Among them, the ISO 9000 quality management system certification is the earliest and most influential; it is of great significance to improve the quality management level of enterprises, which can have significant positive impact on the food safety efforts. HACCP hazard analysis and critical control point management system is unique to the food processing industry to ensure food safety. ISO 22000 food safety management system certification will help food processing companies make better use of the HACCP principle. ISO 22000 will not only address food quality, but also include the establishment of food safety and food safety systems. These management systems or standards form an enterprise’s safety management system from different aspects.

We searched firms’ names from National Certification and Accreditation Information Public Service Platform to find out how many national certifications or accreditations a firm possessed in the year(s) the firm was being studied. We then constructed the variable “food safety certification”, which equals 1 if the firm obtained ISO 22000 certification, ISO 9000 certification, or HACCP certification, and 0 otherwise.

#### 2.1.3. Moderating Variables

The control of the largest shareholder denoted in the model as the variable “first”; for this we take the percentage of common shares owned by the controlling shareholder to measure the extent of the control of the largest shareholder [44]. The ownership of the actual controller of the firm, denoted in the model as the variable SOE, is found from the WIND database. The database is divided into state-owned enterprises if the central or local government equity percentage is greater than or equal to 50 percent. When a firm’s ultimate controlling shareholder is a central or local government entity, the equity property value is 1, otherwise 0 [45].

#### 2.1.4. Control Variables

We included several firm-level control variables to ensure valid results for several reasons: (1) Firm size may influence whether companies possess food safety certification, because different sized companies may exhibit different resource deployment [46]. When the size of the firm is small, the resources will focus more on such immediate needs as cutting costs, rather than on how to fulfill more food safety social responsibility. We controlled and measured the firm size using the common logarithm of the total assets of the firm [44]. (2) The debt ratio captures the effects of resource constraint and creditor power on CSR [47]. Firms consider lenders as a factor when they engage in social activity in China [48]. We used financial leverage, which is interest-bearing debt divided by the total assets, to measure this variable [44]. (3) Following Clarkson, et al. [49] and Du et al. [44], we also consider “Capital intensity”, measured as the ratio of capital spending divided by total sales revenue, which affects voluntary fulfilling social responsibility. (4) Following Du et al. [44], we constructed the indicator variables “CEO” and “Board”. Top executives play an important role in CSR [44], so “CEO” and “Board” were included in our regression models to reflect the influence of top management characteristics. For the measurement of these two variables, we follow Du et al. [44] and define the value of the “CEO” indicator as 1 when the chairman and CEO are the same person and 0 otherwise; and the value of the indicator “Board” is measured by the common logarithm of the number of members on the board of directors. (5) The length of history of an enterprise affects the allocation of its resources, which affects the priority of pursuing food safety certifications. A long-established firm often cares more about its reputation [50] and thus is more willing to apply for food safety certification than a start-up. For this variable, we use “Age”, which is the natural logarithm of the number of years since the firm was established. (6) To control the year effect, we created nine dummy variables based on the year we selected from 2009 to 2017.

### 2.2. Model

Based on the above theoretical analysis and the hypotheses proposed, the model on the relationship between food safety concern in mission statement and food safety certification pursuance suggested in this study is as follows:Food safety certification = α + β_1_ × food safety concern of mission statement + β_2_ × Size + β_3_ × Financial leverage + β_4_ × Capital intensity + β_5_ × CEO + β_6_ × Board + β_7_ × Age + Year Dummies + ε(2)
Food safety certification = α + β1 × food safety concern of mission statement + β2 × First + β_3_ × food safety concern of mission statement × First + β_4_ × Size + β_5_ × Financial leverage + β_6_ × Capital intensity + β_7_ × CEO + β_8_ × Board + β_9_ × Age + Year Dummies + ε(3)
Food safety certification = α + β1 × food safety concern of mission statement + β2 × First + β_3_ × food safety concern of mission statement × First + β_4_ × Size + β_5_ × Financial leverage + β_6_ × Capital intensity + β_7_ × CEO + β_8_ × Board + β_9_ × Age + Year Dummies + ε(4)

## 3. Results

Table 1 presents the descriptive statistics for the model variables. We found that the maximum value in the food safety concern of mission statement is 1. The mission statement in that sample is “Passed ISO 9001 international quality system certification and HACCP food safety control system certification”. The average value of food safety concern of mission statement is 0.07 and the standard deviation is 0.08. We also found that 95 companies (out of the 115 companies in our sample) have applied for food safety certification for their products, and the average value in the food safety certification is 0.63, which means that 63% of food processing firms possess food safety certification for their products. In order to explore the relationship between variables, we used a Pearson correlation coefficient to test the correlation between variables.

Table 2 shows the results of empirical inspections of enterprises possession of food safety certification. Among them, Model 1 is the regression result of zero models, and regression analysis is performed on all control variables involved in the model. The regression results of Model 2 show that the regression coefficient of food safety concern in mission statement is 3.045, which is significant at the level of *p* < 0.05. This result indicates that the higher the mission statement’s focus on food safety, the stronger the company’s commitment to obtaining food safety certification. Hypothesis 1 has been supported.

Subsequently, to examine the moderating effect of the control of the largest shareholder (variable First), we found a significant interaction between mission statement and control of First (β = 0.411, *p* < 0.01, in Table 2, Model 3). This result means that compared to firms with more distributed shareholder ownership, in firms with higher portion of equity concentrated on the largest stakeholder, the positive correlation between food safety concerns and possession of food safety certification would be strengthened. Hypothesis 2 has been supported. 

In the above subsections we examined the hypotheses on the relationship between food safety concerns (as appearing in mission statements) and the possession of food safety certifications of the involved food processing firms, as well as the moderating roles of the concentration of shareholder ownership in the above relationship. Now we examine the role of ownership nature (state-owned or non-state-owned) in the relationship between food safety concern in mission statement (the intended goal) and food safety certification possession (the realized outcome). Hypothesis 3 predicted that state ownership would strengthen the effect of the food safety concern in mission statement on the possession of food safety certification. Our findings confirmed that coefficient for food safety concern of mission statement × SOE is 0.637, significant at *p* < 0.01 (in Table 2, Model 4). That is, compared to non-state-owned enterprises, the positive correlation between food safety concerns and possession of food safety certification will be strengthened in state-owned enterprises. Hypothesis 3 has been supported.

We further plotted the moderating effects of largest shareholder (the variable First) and SOE. We normalized the independent variable food safety concern of mission statement, the dependent variable food safety certification, and all control variables in models 3 and 4 and then perform the regression [51]. Based on the regression results, we plotted the corresponding adjustment effect diagram (Figure 2) to display the effect of food safety concern of mission statement on food safety certification at different levels of largest shareholder. It shows that the effect of food safety concern of mission statement is stronger for higher shareholding of the largest shareholder than lower shareholding of the largest shareholder.

In order to further verify the moderating role of the largest shareholder’s degree of control, we used a sub-sample test method. By taking the calculated moderation controlling the proportion of largest shareholders as the dividing point, this study divided the 872 observations into two groups: high shareholder control, or Model 1 (those with the value of the variable First above the mean) and low shareholder control, or Model 2 (those with value of variable First below the mean). These two groups of samples were then regressed separately, and the results are shown in Table 3. 

It can be seen from Table 3 that the coefficient of the food safety concern of mission statement of model 1 is 7.336, which is significantly higher than the coefficient of the food safety concern of mission statement of model 2 (1.986), and it is very significant (*p* < 0.01). Hypothesis 2 has also been supported

At the same time, to further test hypothesis 3, we divided the 872 observations into two groups: SOEs and non-SOEs. These two groups of samples were then regressed separately: Model 1 with SOEs, and Model 2 with non-SOEs. The results are shown in Table 4. It can be seen from Table 4 that the coefficient of the food safety concern of mission statement for model 1 is 16.693 and significant (*p* < 0.01), while the same coefficient (food safety concern of mission statement) for model 2 is not significant. Hypothesis 3 has also been supported

Finally, to test the relationship between food safety concern in mission statement and food safety certification and the moderating effects of the control of first shareholder and state ownership, we employed an ordinary least squares (OLS) model and used firm fix effects and industry/year fixed effects in the model estimation. We report the results in Table 5 The findings are similar to those in our primary testing: the interaction term between food safety concern in mission statemen and the control of first shareholder in model 2 has a positive effect on food safety certification, supporting Hypothesis 2; the interaction term between food safety concern in mission statement and state ownership in model 3 also has a positive effect on food safety certification, supporting Hypothesis 3.

## 4. Discussion

Based on the data of the publicly listed food industry companies in China from 2009 to 2018, we found that the companies whose mission statements paid close attention to food safety are more likely to possess food safety certifications; in other words, a food processing company’s food safety focus in its mission statement helps the company to pursue and fulfil its food safety social responsibility. In terms of equity distribution, it is found that the firms whose largest shareholder controls a substantially large percentage of shares tend to have stronger positive relationship between the mission statement food safety concern and the food safety certification possession, compared to those firms where equity distribution is more diluted or scattered. Moreover, in state-owned enterprises, the positive correlation between food safety concerns and possession of food safety certification is stronger compared to non-state-owned enterprises.

This study contributes to and expands past research in several ways. First, our study supplements research on corporate food safety social responsibility. Previous studies have mostly considered the concern of food safety and corporate social responsibility from the perspective of agency theory [1,2]. We have found that it is possible to increase the focus on food safety information in mission statements to gain legitimacy and thereby motivate and guide companies to fulfill their social responsibility on food safety. Second, scholars proved the effects of mission statement on firms’ performance and employees’ behavior [11,50,51,52,53,54]. However, they did not notice the relationship between the mission statement and the company’s CSR-related strategy or actions. According to legitimacy theory, we view the mission statement as an internal norm and an external cognitive of the enterprise. Mission statement can legitimize the organization’s behavior of pursuing food safety responsibility. Under the guidance of the mission statement and the pressure from the normative legitimacy of the mission statement, companies would be more actively pursuing food safety certifications. Finally, the results of this paper also provide a general logic for explaining the impact of food safety attention on corporate social responsibility activities under different shareholding nature. Under different equity natures, the pressures on the regulatory legitimacy of the firm are different and the impact of mission statement on the possession of food safety certification is also different.

On the other hand, we recognize that the current study also has some limitations. First, in this study, mission statements were assessed using what has been considered in the literature the concerns about food safety. Future research should explore approaches such as analyzing the presence of different types of stakeholders in the mission statements [17] or using social network analytics to measure the related concerns in firms’ missions [20]. Second, in some companies employees also participate in the formulation of mission statements; in addition, company missions are likely to change with governmental policy changes, especially in the context of Chinese business environment. Future research could place more emphasis on the impact of employee participation, and on the impact of governmental policies on the characteristics of mission statements. Moreover, the nature of shareholders was revealed to be an essential moderating variable in our research. In that direction, future studies should assess the mediating role of other relevant constructs, such as organizational commitment [55].

## 5. Conclusions

In this study, we provide a detailed examination, from internal and external perspectives, of the relationship between the mission statements of food processing companies and the possession of food safety certification by those companies. A mission statement is a company’s declaration of and commitment to its own goals and the fundamental paths to achieve the goals. As the compass of the company’s future development, the mission statement helps the company build internal norms and supervise the implementation of the established norms. At the same time, as an important part of the company’s external recognition, as companies leverage mission statements to establish corporate identity. Under the guidance of a clear identity positioning, the company’s business activities will reinforce the public’s cognition of the company’s image, thus maintaining the company’s identity and reputation. Under the pressure of normative legitimacy and culture-cognitive legitimacy, firms whose mission statement concerns more about food safety tend to possess food safety certification. Moreover, we identified the moderating role of the equity ratio of largest shareholder in the relationship between mission statement food safety concern and food safety certification possession. Finally, our results suggest that the association of the safety concern in corporate mission and food safety certification possession for SOEs is more positive than that for non-SOEs, as SOEs assume greater responsibility of maintaining food safety social responsibility.

## Figures and Tables

**Figure 1 ijerph-17-04735-f001:**
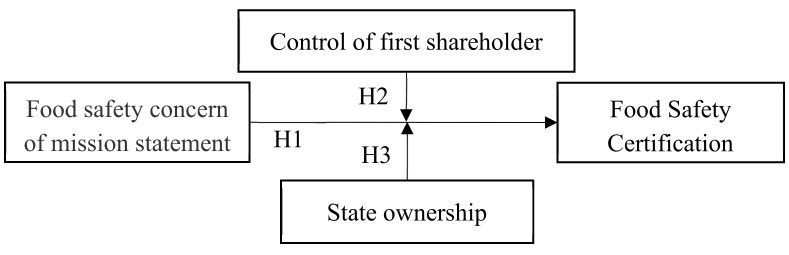
A theoretical model. H1, H2, H3 mean Hypotheses 1–3.

**Figure 2 ijerph-17-04735-f002:**
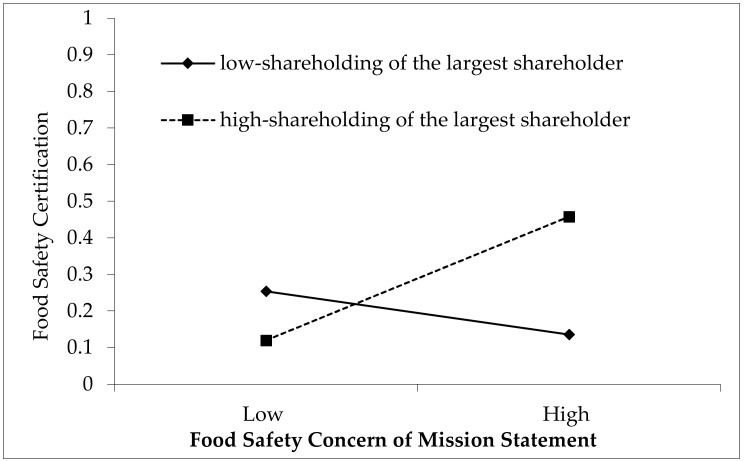
The moderating effect of the control of the largest shareholder on the relationship between the Food Safety Concern of Mission Statement and Food Safety Certification.Figure 3. exhibits the effect of food safety concern of mission statement on food safety certification depending on whether the company concerned are SOEs or non-SOEs. Consistent with our prediction, the impact of food safety concern of mission statement is significant and strong for SOEs, while for non-SOEs the relationship is not statistically significant.

**Figure 3 ijerph-17-04735-f003:**
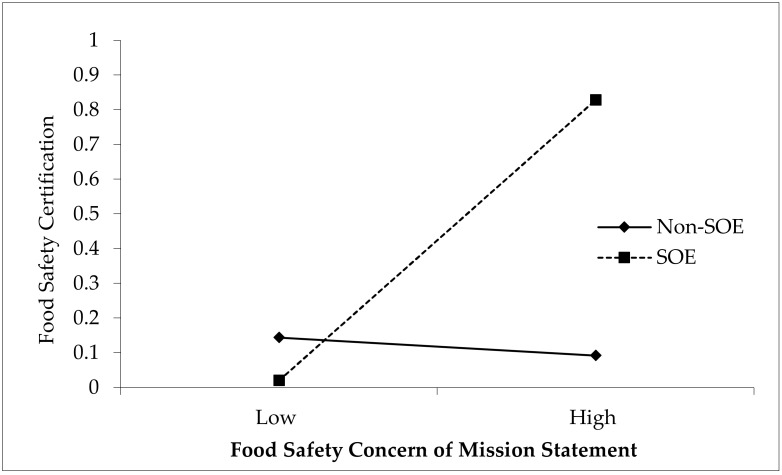
The moderating effect of state ownership on the relationship between the food safety concern of mission statement and food safety certification.

**Table 1 ijerph-17-04735-t001:** Descriptive statistics and correlations.

Variables	Min	Max	Mean	S.D.	1	2	3
1. Food safety certification	0.00	1.00	0.63	0.48	-		
2. Food safety concern of mission statement	0.00	1.00	0.07	0.08	0.09 **	-	
3. SOE (d)	0.00	1.00	0.43	0.50	0.13 **	−0.06	-
4. First	0.05	0.96	0.38	0.15	0.10 **	0.00	0.15 **
5. Size	19.24	25.80	21.96	1.06	0.18 **	−0.01	0.15 **
6. CEO (d)	0.00	1.00	0.26	0.44	−0.01	0.03	−0.26 **
7. Board	0.14	0.80	0.38	0.07	−0.02	0.01	0.04
8. Financial leverage	0.02	1.51	0.37	0.19	−0.10 **	0.03	0.08 *
9. Capin	0.18	357.29	2.52	12.75	−0.07 *	−0.01	−0.05
10. Age	0.00	3.50	2.78	0.33	0.06	−0.08 *	0.12 **
	3	4	5	6	7	8	9
3. SOE	-						
4. First	0.15 **	-					
5. Size	0.15 **	0.16 **	-				
6. CEO (d)	−0.26 **	0.01	−0.05	-			
7. Board	0.04	0.08 *	0.11 **	0.00	-		
8. Lev	0.08 *	−0.07 *	0.20 **	−0.13 **	0.03	-	
9. Capin	−0.05	−0.07 *	−0.06	−0.04	−0.05	0.07 *	-
10. Age	0.12 **	−0.14 **	0.14 **	0.02	0.06	−0.04	0.00

*n* = 872; * *p* < 0.05; ** *p* < 0.01 (two-tailed); (d): dummy variable; We define the value of the “CEO” indicator as 1 when the chairman and CEO are the same person and 0 otherwise; SOE: state-owned enterprise.

**Table 2 ijerph-17-04735-t002:** The Empirical Results.

Dependent Variable	Food Safety Certification				
	Model 1	Model 2	Model 3	Model 4	Model 5
Size	0.355 *** (0.080)	0.318 *** (0.084)	0.370 *** (0.087)	0.263 *** (0.085)	0.335 *** (0.089)
CEO	−0.269 (0.171)	−0.113 (0.178)	−0.025 (0.182)	−0.125 (0.180)	0.013 (0.186)
Board	−1.757 (1.116)	−2.169 * (1.155)	−2.509 ** (1.174)	−2.747 ** (1.210)	−3.104 ** (1.235)
Financial leverage	−1.208 *** (0.431)	−1.269 *** (0.445)	−1.196 *** (0.449)	−1.467 *** (0.455)	−1.460 *** (0.460)
Capin	−0.042 * (0.022)	−0.033 (0.021)	−0.034 (0.022)	−0.033 (0.021)	−0.034 (0.022)
Age	−0.334 (0.252)	−0.407 (0.264)	−0.339 (0.263)	−0.313 (0.262)	−0.217 (0.261)
Year	Included	Included	Included	Included	Included
First		0.626 (0.543)	0.827 (0.554)	1.095 ** (0.557)	1.435 ** (0.576)
SOE		0.669 *** (0.166)	0.743 *** (0.170)	0.852 *** (0.182)	0.960 *** (0.185)
Food safety concern of mission statement		3.045 ** (1.257)	3.627 *** (1.390)	6.537 *** (1.336)	6.147 *** (1.365)
Food safety concern of mission statement X First			0.411 *** (0.129)		0.479 *** (0.126)
Food safety concern of mission statement X SOE				0.637 *** (0.127)	0.720 *** (0.130)
Constant	−4.485 ** (1.895)	−3.988 ** (1.954)	−5.410 *** (2.038)	−3.136 (1.986)	−5.045 ** (2.083)
Estimation model	Logistic	Logistic	Logistic	Logistic	Logistic
Number of observations	872	872	872	872	872
Log Likelihood	−532.219	−518.815	−513.562	−504.247	−496.708

Standard errors are in parentheses; Interaction items are standardized. * *p* < 0.1; ** *p* < 0.05; *** *p* < 0.01.

**Table 3 ijerph-17-04735-t003:** First shareholder control degree grouping regression results.

Dependent Variable	High Shareholder Control	Low Shareholder Control
	Food Safety Certification	
	Model 1	Model 2
Size	0.505 *** (0.137)	0.246 ** (0.111)
CEO	0.379 (0.261)	−0.539 * (0.278)
Board	−2.868 * (1.584)	−3.097 * (1.868)
Financial leverage	−1.466 ** (0.643)	−1.399 ** (0.649)
Capin	−0.218 * (0.116)	−0.034 (0.022)
Age	0.353 (0.321)	−1.787 *** (0.519)
Year	Included	Included
SOE	0.710 *** (0.257)	0.614 ** (0.243)
Food safety concern of mission statement	7.336 *** (2.181)	1.986 (1.889)
Constant	−9.912 *** (3.102)	2.535 (2.843)
Estimation model	Logistic	Logistic
Number of observations	439	433
Log Likelihood	−246.270	−256.799

Standard errors are in parentheses. * *p* < 0.1; ** *p* < 0.05; *** *p* < 0.01.

**Table 4 ijerph-17-04735-t004:** Ownership Nature Grouping Regression Results.

Dependent Variable	SOE = 1	SOE = 0
	Food Safety Certification	
	Model 1	Model 2
Size	0.476 *** (0.145)	0.088 (0.116)
CEO	1.195 ** (0.497)	−0.461 ** (0.211)
Board	−2.531 (1.793)	−2.331 (1.790)
Financial leverage	−2.840 *** (0.841)	−0.789 (0.581)
Capin	0.380 ** (0.166)	−0.048 * (0.027)
Age	0.039 (0.555)	−0.269 (0.320)
Year	Included	Included
First	0.911 (1.191)	1.458 ** (0.682)
Food safety concern of mission statement	16.693 *** (3.232)	−0.264 (1.003)
Constant	−9.665 *** (3.684)	0.977 (2.752)
Estimation model	Logistic	Logistic
Number of observations	377	495
Log Likelihood	−183.535	−303.818

Standard errors are in parentheses. * *p* < 0.1; ** *p* < 0.05; *** *p* < 0.01.

**Table 5 ijerph-17-04735-t005:** The Robust Test.

Dependent Variable	Food Safety Certification			
	Model 1	Model 2	Model 3	Model 4
Size	0.061 *** (0.016)	0.067 *** (0.016)	0.055 *** (0.016)	0.062 *** (0.016)
CEO	−0.017 (0.037)	−0.001 (0.038)	−0.021 (0.037)	0.001 (0.037)
Board	−0.467 * (0.238)	−0.504 ** (0.238)	−0.586 ** (0.236)	−0.655 *** (0.235)
Lev	0.231 *** (0.089)	−0.218 ** (0.089)	−0.270 *** (0.088)	−0.255 *** (0.088)
Capin	−0.002 (0.001)	−0.002 (0.001)	−0.002 (0.001)	−0.002 (0.001)
Age	−0.086 (0.054)	−0.073 (0.054)	−0.064 (0.054)	−0.043 (0.054)
Year	Included	Included	Included	Included
First	0.144 (0.112)	0.157 (0.112)	0.231 ** (0.112)	0.262 ** (0.112)
SOE	0.141 *** (0.034)	0.150 *** (0.034)	0.149 *** (0.034)	0.163 *** (0.034)
Food safety concern of mission statement	0.456 ** (0.186)	0.313 (0.195)	1.078 *** (0.220)	0.948 *** 0.222)
Food safety concern of mission statement X First		0.055 ** (0.024)		0.080 *** (0.024)
Food safety concern of mission statement X SOE			0.104 *** (0.020)	0.117 *** (0.020)
Estimation model	OLS	OLS	OLS	OLS
Number of observations	872	872	872	872
*R^2^*	0.067	0.073	0.095	0.107

Standard errors are in parentheses; Interaction items are standardized. * *p* < 0.1; ** *p* < 0.05; *** *p* < 0.01; OLS means ordinary least squares.

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
