# Peer review of "How Does Mission Statement Relate to the Pursuit of Food Safety Certification by Food Companies?"

_ijerph, 2020, doi:10.3390/ijerph17134735_

Round 1
Reviewer 1 Report
The food safety efforts of Chinese companies are a very important issue for also those of neighboring countries. I feel this paper, which uses the data to analyze this point, is very interesting. However, because of China's policy idiosyncrasies, I feel that there are many aspects of China that readers around the world, especially except of china, can not understand, such as the extent to which Chinese companies are required to disclose information about their companies under the Chinese domestic system such as mission statements. First of all, I think it needs to be explained in such a way that those readers can understand the relevant institutional design of China.
Other points that should be modified are as follows.
Page 5 Line 189
The sample is a very important point in this study. At the very least, the data used in the analysis should be disclosed in the Supplement, etc. Without disclosure, it is not possible to determine whether this result is correct.
Page 7 Line 287
Is the statement “Passed ISO 9001 international quality system certification and HACCP food safety control system certification” can be regarded as a mission statement? This includes gaining HACCP certification in it, complicating relationship with food safety certification. I think that variables directly related to food certification should be excluded from the independent variables.
Page 10 Line 338
It is stated “Hypothesis 3 has also been supported” but testing two times for one hypothesis should be avoided. More robust setting should be considered primary testing, and the secondary testing should be regarded as supportive.
Author Response
Thank you for your suggestion. We have added some information on Page 5 Line 205-215. We have added the information that the China Securities Regulatory Commission (CSRC) requires listed companies to disclose in their annual reports. Although CSRC did not require that the company must disclose corporate mission statement information, the importance of mission statements is promoted by the Forum of Chinese Management Thoughts jointly launched in 2001 by Tsinghua University, Peking University and several other leading universities and consulting firms. The Forum is committed to helping top Chinese business executives to simplify complex management theories and localize western corporate cultural values.
Other points that should be modified are as follows.
Page 5 Line 189
The sample is a very important point in this study. At the very least, the data used in the analysis should be disclosed in the Supplement, etc. Without disclosure, it is not possible to determine whether this result is correct.
Thank you for your suggestion. We have compiled our sample table for this data analysis in data.xlsx
Page 7 Line 287
Is the statement “Passed ISO 9001 international quality system certification and HACCP food safety control system certification” can be regarded as a mission statement? This includes gaining HACCP certification in it, complicating relationship with food safety certification. I think that variables directly related to food certification should be excluded from the independent variables.
Thank you for your suggestion. Pearce (1982) argued that the mission statement also includes the corporate vision and values. The company's vision represents its goals for future development. We consider that a company mentioned HACCP in its mission statement is to tell investors and consumers its determination to put food safety on high priority.
We also considered your suggestion and reran the data using the mission statement without certification in it. For H1, the result shows that the regression coefficient of food safety concern in mission statement is 3.12, which is significant at the level of p < 0.05. For H2, we found a significant interaction between mission statement and control of First (β=0.41, p<0.01). For H3, our findings confirmed that coefficient for food safety concern of mission statement × SOE is 0.53, significant at p<0.01.
To sum up, regardless of whether or not we include "obtaining food safety certification" in the mission statement, the research results support our hypotheses.
Page 10 Line 338
It is stated “Hypothesis 3 has also been supported” but testing two times for one hypothesis should be avoided. More robust setting should be considered primary testing, and the secondary testing should be regarded as supportive.
Thank you for your suggestion. We have adjusted the article structure and moved the support test to the robust test section.
Reviewer 2 Report
This paper is a straightforward and simple analysis which finds that the inclusion of food safety language in a Chinese company’s mission statement is associated with food safety certification. The authors also find evidence that companies owned by a consolidated shareholder or state-owned enterprises are more likely to have at least one food safety certification, with an even stronger effect when either consolidated shareholders or state-owned enterprises have mission statements that include food safety language.
The analysis of this paper may be adequate for the intended hypothesis testing. However, it seems that the authors have avoided performing some analyses that could provide more specificity to their findings. I identify one main shortcoming and one unresolved question:
- Why do the authors not use random effects in their panel data model? In the U.S. sectors I’m familiar with, the use of food safety certifications vary by commodity or by marketing channel, with commodities or channels that have had food safety issues in the past to be associated with the use of certifications in the present. Using the example the authors cite of melamine-contaminated infant formula, after 2008 I’d expect that food manufacturers of baby foods were likely obtaining certifications at a higher rate than manufacturers of other foods. Firm-level random effects would account for this and is the major benefit of using panel data.
- Do the authors have mission statements that vary across time? Do the authors have the date at which the mission statements were written? I don’t have any information, but I would guess the authors have mission statements from a single point in time. Please clarify in the data description.
The English in this paper is understandable and, for the most part, grammatically correct. However, some terms are vague and some sentences do not convey clear arguments. See examples from the abstract, comments in italics, suggested revisions in bold:
Food safety has long been a major public concern in China. One [question] of the food processing industry's social responsibility is whether a [food processing] company should pursue food safety certification for its products. (This sentence is grammatically incorrect without the word “question” and confusing with the extra qualifying statements.)
For [As part of] their corporate image, some food processing companies focus on food safety in their corporate mission statements. [To enhance the legitimacy of a] mission statement, as a guide for a firm, can provide food companies the legitimacy of [may] pursue food safety certification. This sentence is very confusing as worded. The mission statement doesn’t give the firm permission, support, or legitimacy to get the certification. The certification provides legitimacy for the statement.
However, we find that under different [concentrations of shareholder ownership] equity natures, the pressures on the normative legitimacy of the firm are different and the impact of [food safety language] in mission statement[s] on the possession of food safety certification[s] is also differ[s]ent. The term “equity natures” was very confusing to me. It wasn’t until later in the paper that I realized the authors were referring to the financial term “equity” and not “equal measures of certain characteristics”. Additional language about “normative legitimacy” muddles the argument and confused me.
By analyzing the mission statement of companies in [the] Chinese food industry, we find that firms with a mission focusing on food safety concerns and on consumers are more willing to pursue food safety certification. I didn’t see consumers being part of your measure of language in the methods section.
Moreover, compared [to]with the firms with relatively scattered equity[more distributed shareholder ownership], in firms where the largest shareholder has substantial domination, the relationship between mission statements and the possession of food safety certification is stronger; compared [to]with non-state-owned enterprises, in state-owned enterprise (SOEs), the relationship between firm mission statements of and the acquisition of food safety certification is stronger. To me, the term “scattered equity” implies small pockets of equal measures in random places.
Author Response
Thank you for your suggestion. The random effect model could facilitate generalization to the population represented by the samples. In the context of Chinese businesses, compared to American companies, majority of Chinese companies have not yet fully understood the important role of mission statements. Many unlisted (non-pubic) SMEs do not understand the purpose and significance of mission statements. Compared to those smaller companies, the listed companies at least have a clearer vision of their future, which helps to make sense for studying their mission statements. The purpose of our research is to study the role of the mission statement containing the stated intent to pursue food safety certification, and our samples covered all listed food companies (except a few whose data were incomplete). With such a scope and such a treatment, it is more suitable for us to use the fixed effect model.
Although the random effect model might seem effective, the fixed effect part placed in the random term would introduce endogenous problems. Therefore, considering the endogenous issue, it is better to use a fixed-effect model. We supplemented the robust detection of fixed-effect models in the article. The results still support our hypothesis.
Last, because the corporate mission represents the reason for its existence and development direction, very few companies will frequently change their mission statements. The mission statements we selected were collected October 2019. We have added related Information on Page 5 Line 213-215
The English in this paper is understandable and, for the most part, grammatically correct. However, some terms are vague and some sentences do not convey clear arguments. See examples from the abstract, comments in italics, suggested revisions in bold:
Food safety has long been a major public concern in China. One [question] of the food processing industry's social responsibility is whether a [food processing] company should pursue food safety certification for its products. (This sentence is grammatically incorrect without the word “question” and confusing with the extra qualifying statements.)
Thank you for your suggestion. We have corrected it on Page 1 Line 11-12
For [As part of] their corporate image, some food processing companies focus on food safety in their corporate mission statements. [To enhance the legitimacy of a] mission statement, as a guide for a firm, can provide food companies the legitimacy of [may] pursue food safety certification. This sentence is very confusing as worded. The mission statement doesn’t give the firm permission, support, or legitimacy to get the certification. The certification provides legitimacy for the statement.
Thank you for your question. Pearce (1982) shows that the mission statement also includes the corporate vision and values. The concern about food safety issues in the mission statement can be seen as the goal to be pursued by the enterprise. In this respect, the mission statement is not only a statement but also a signal of the extent to which the firm practices its mission statement (Bart, 1997; Bartkus et al., 2008; Suh et al., 2011). The mission statement explains the ‘why’ of the firm, expressed in terms of the firm’s operational processes and procedures that form the dynamism of mission implementation (Rey et al., 2017). This mission statement (language concern about food safety issues) can give the management support and legitimacy to obtain certification for food safety. We have added your suggestion on Page 1 Line 13-16
However, we find that under different [concentrations of shareholder ownership] equity natures, the pressures on the normative legitimacy of the firm are different and the impact of [food safety language] in mission statement[s] on the possession of food safety certification[s] is also differ[s]ent. The term “equity natures” was very confusing to me. It wasn’t until later in the paper that I realized the authors were referring to the financial term “equity” and not “equal measures of certain characteristics”. Additional language about “normative legitimacy” muddles the argument and confused me.
Thank you for your suggestion. We have corrected it on Page 1 Line 16-19
By analyzing the mission statement of companies in [the] Chinese food industry, we find that firms with a mission focusing on food safety concerns and on consumers are more willing to pursue food safety certification. I didn’t see consumers being part of your measure of language in the methods section.
Thank you for your question. We have deleted it on Page 1 Line 11-12
Moreover, compared [to]with the firms with relatively scattered equity[more distributed shareholder ownership], in firms where the largest shareholder has substantial domination, the relationship between mission statements and the possession of food safety certification is stronger; compared [to]with non-state-owned enterprises, in state-owned enterprise (SOEs), the relationship between firm mission statements of and the acquisition of food safety certification is stronger. To me, the term “scattered equity” implies small pockets of equal measures in random places.
Thank you for your suggestion. We have corrected it on Page 1 Line 20-23